# Challenging Common Assumptions in Convex Reinforcement Learning

**Mirco Mutti**\*
Politecnico di Milano
Università di Bologna
mirco.mutti@polimi.it

**Riccardo De Santi**\*
ETH Zurich
rdesanti@ethz.ch

**Piersilvio De Bartolomeis**
ETH Zurich
pdebartol@ethz.ch

**Marcello Restelli**
Politecnico di Milano
marcello.restelli@polimi.it

## Abstract

The classic Reinforcement Learning (RL) formulation concerns the maximization of a *scalar* reward function. More recently, *convex* RL has been introduced to extend the RL formulation to all the objectives that are convex functions of the state distribution induced by a policy. Notably, convex RL covers several relevant applications that do not fall into the scalar formulation, including imitation learning, risk-averse RL, and pure exploration. In classic RL, it is common to optimize an *infinite trials* objective, which accounts for the state distribution instead of the empirical state visitation frequencies, even though the actual number of trajectories is always finite in practice. This is theoretically sound since the infinite trials and finite trials objectives are equivalent and thus lead to the same optimal policy. In this paper, we show that this hidden assumption does not hold in convex RL. In particular, we prove that erroneously optimizing the infinite trials objective in place of the actual *finite trials* one, as it is usually done, can lead to a significant approximation error. Since the finite trials setting is the default in both simulated and real-world RL, we believe shedding light on this issue will lead to better approaches and methodologies for convex RL, impacting relevant research areas such as imitation learning, risk-averse RL, and pure exploration among others.

## 1 Introduction

Standard Reinforcement Learning (RL) [50] is concerned with sequential decision-making problems in which the utility can be expressed through a linear combination of scalar reward terms. The coefficients of this linear combination are given by the state visitation distribution induced by the agent's policy. Thus, the objective function can be equivalently written as the inner product between the mentioned state distribution and a reward vector. However, not all the relevant objectives can be encoded through this linear representation [2]. Several works have thus extended the standard RL formulation to address non-linear objectives of practical interest. These include imitation learning [30, 42], or the problem of finding a policy that minimizes the distance between the induced state distribution and the state distribution provided by experts' interactions [1, 29, 32, 33, 22, 17], risk-averse RL [20], in which the objective is sensitive to the tail behavior of the agent's policy [52, 43, 51, 16, 15, 8, 61], pure exploration [27], where the goal is to find a policy that maximizes the entropy of the induced state distribution [53, 33, 41, 40, 59, 24, 36, 47, 55, 39, 38], diverse skills

---

\*Equal contribution

36th Conference on Neural Information Processing Systems (NeurIPS 2022).

discovery [23, 19, 25, 48, 11, 35, 28, 58], constrained RL [4, 3, 9, 37, 45, 56, 6], and others. All this large body of work has been recently unified into a unique framework, called *convex* RL [60, 57, 21], which admits as an objective any convex function of the state distribution induced by the agent's policy. The convex RL problem has been showed to be largely tractable either computationally, as it admits a dual formulation akin to standard RL [44], or statistically, as principled algorithms achieving sub-linear regret rates that are slightly worse than standard RL have been developed [60, 57].

However, we note that the usual convex RL formulation makes an implicit *infinite trials* assumption which is rarely met in practice. Indeed, the objective is written as a function of the state distribution, which is an expectation over the empirical state distributions that are actually obtained by running the policy in a given episode. In practice, we always run our policy for a finite number of episodes (or *trials*), which in general prevents the empirical state distribution from converging to its expectation. This has never been a problem in standard RL: due to the scalar objective, optimizing the policy over infinite trials or finite trials is equivalent, as it leads to the same

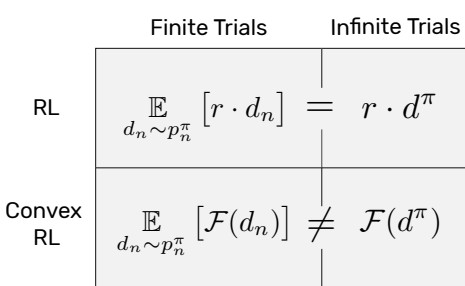

Figure 1: Summary of the main finding of this paper: the equivalence between finite and infinite trials objectives does not hold for the convex RL formulation.

optimal policy. Crucially, in this paper, we show that this property does not hold for the convex RL formulation: a policy optimized over infinite trials can be significantly sub-optimal when deployed over finite trials (Figure 1). In light of this observation, we reformulate the convex RL problem from a *finite trials* perspective, developing insights that can be used to partially rethink the way convex objectives have been previously addressed in RL, with potential ripple effects to research areas of significant interest, such as imitation learning, risk-averse RL, pure exploration, and others.

In this paper, we formalize the notion of a finite trials RL problem, in which the objective is a function of the empirical state distribution induced by the agent's policy over $n$ trials rather than its expectation over infinite trials. As an illustrative example, consider a financial application, in which we aim to optimize a trading strategy. In the real world, we can only deploy the strategy over a single trial. Thus, we are only interested in the performance of the strategy in the real-world realization, rather than the performance of the strategy when averaging different realizations. Similar considerations apply to other relevant real-world applications, such as autonomous driving or treatment optimization in a healthcare domain. Following this intuition, we first define the (linear) finite trial RL formulation (Section 3), for which it is trivial to prove the equivalence with standard RL. In Section 4, we provide the finite trial convex RL formulation, for which we prove an upper bound on the approximation error made by optimizing the infinite trials as a proxy of the finite trials objective. In light of this finding, we challenge the hidden assumption that **(1)** *convex RL can be equivalently addressed with an infinite trials formulation*, even if the setting is finite-trial. We corroborate this result with an additional numerical analysis showing that the approximation bound is non-vacuous for relevant applications (Section 6). Finally, in Section 5 we include an in-depth analysis of the *single trial* convex RL, which suggests that other common assumptions in the convex RL literature, i.e., that **(2)** *the problem is always computationally tractable* and that **(3)** *stationary policies are in general sufficient*, should be reconsidered as well. The proofs of the reported results can be found in the Appendix.

## 2 Preliminaries

In this section, we report the notation and background useful to understand the paper. We denote with $[T]$ a set of integers $\{1, \ldots, T\}$, and with a lower case letter $a$ a scalar or a vector, according to the context. For two vectors $a, b \in \mathbb{R}^d$, we denote with $a \cdot b = \sum_{i=1}^d a_i b_i$ the inner product between $a, b$.

### 2.1 Probabilities and Percentiles

Let $\mathcal{X}$ denote a measurable space, we will denote with $\Delta(\mathcal{X})$ the probability simplex over $\mathcal{X}$, and with $p \in \Delta(\mathcal{X})$ a probability measure over $\mathcal{X}$. For two probability measures $p, q$ over $\mathcal{X}$, we define their $\ell^p$-distance as $\|p - q\|_p := \left( \sum_{x \in \mathcal{X}} |p(x) - q(x)|^p \right)^{1/p}$, and their Kullback-Leibler (KL) divergence

as $\mathrm{KL}(p||q) := \sum_{x \in \mathcal{X}} p(x) \log \big(p(x)/q(x)\big)$. Let $X$ be a random variable distributed according to $p$, having a cumulative density function $F_X(x) = Pr(X \leq x)$. We denote with $\mathbb{E}[X]$ its expected value, and its $\alpha$-percentile is denoted as $\mathrm{VaR}_\alpha(X) = \inf \big\{x \mid F_X(x) \geq \alpha\big\} = F_X^{-1}(\alpha)$, where $\alpha \in (0, 1)$ is a confidence level, and $\mathrm{VaR}_\alpha$ stands for Value at Risk (VaR) at level $\alpha$. We denote the expected value of $X$ within its $\alpha$-percentile as $\mathrm{CVaR}_\alpha(X) = \mathbb{E}\big[X \mid X \leq \mathrm{VaR}_\alpha(X)\big]$, where $\mathrm{CVaR}_\alpha$ stands for Conditional Value at Risk (CVaR) at level $\alpha$.

## 2.2 Markov Decision Processes

A tabular Markov Decision Process (MDP) [44] is defined as $\mathcal{M} := (\mathcal{S}, \mathcal{A}, P, T, \mu, r)$, where $\mathcal{S}$ is a state space of size $S$, $\mathcal{A}$ is an action space of size $A$, $P$ is a Markovian transition model $P : \mathcal{S} \times \mathcal{A} \to \Delta(\mathcal{S})$, such that $P(s'|s, a)$ denotes the conditional probability of the next state $s'$ given the current state $s$ and action $a$, $T$ is the episode horizon, $\mu$ is an initial state distribution $\mu : \Delta(\mathcal{S})$, and $r$ is a scalar reward function $r : \mathcal{S} \to \mathbb{R}$, such that $r(s)$ is the reward collected in the state $s$.

In the typical interaction episode, an agent first observes the initial state $s_0 \sim \mu$ of the MDP. Then, the agent select an action $a_0$, so that the MDP transitions to the next state $s_1 \sim P(\cdot|s_0, a_0)$, and the agent collects the reward $r(s_1)$. Having observed $s_1$, the agent then selects an action $a_1$ triggering a subsequent transition to $s_2 \sim P(\cdot|s_1, a_1)$. This process carries on repeatedly until the episode ends.

A policy $\pi$ defines the behavior of an agent interacting with an MDP, i.e., the strategy for which an action is selected at any step of the episode. It consists of a sequence of decision rules $(\pi_t)_{t=0}^\infty$ that maps the current trajectory[2] $h_t = (s_i, a_i)_{i=0}^{t-1} \in \mathcal{H}_t$ with a distribution over actions $\pi_t : \mathcal{H}_t \to \Delta(\mathcal{A})$, where $\mathcal{H}_t$ denotes the set of trajectories of length $t$. A non-stationary policy is a sequence of decision rules $\pi_t : \mathcal{S} \to \Delta(\mathcal{A})$. A stationary (Markovian) policy is a time-consistent decision rule $\pi : \mathcal{S} \to \Delta(A)$, such that $\pi(a|s)$ denotes the conditional probability of taking action $a$ in state $s$.

A trajectory $h$, obtained from an interaction episode, induces an empirical distribution $d$ over the states of the MDP $\mathcal{M}$, such that $d(s) = \frac{1}{|h|} \sum_{s_t \in h} \mathbb{1}(s_t = s)$. We denote with $p^\pi$ the probability of drawing $d$ by following the policy $\pi$. For $n \in \mathbb{N}$, we denote with $d_n$ the empirical distribution $d_n(s) = \frac{1}{n} \sum_{i=1}^n d_i(s)$, and with $p_n^\pi$ the probability of drawing $d_n$ by following the policy $\pi$ for $n$ episodes. Finally, we call the expectation $d^\pi = \mathbb{E}_{d \sim p^\pi}[d]$ the state distribution induced by $\pi$.

# 3 Reinforcement Learning in Finite Trials

In the standard RL formulation [50], an agent aims to learn an optimal policy by interacting with an unknown MDP $\mathcal{M}$. An optimal policy is a decision strategy that maximizes the expected sum of rewards collected during an episode. Especially, we can represent the value of a policy $\pi$ through the value function $V_t^\pi(s) := \mathbb{E}_\pi \big[ \sum_{t'=t}^T r(s_{t'}) \mid s_t = s \big]$. The value function allows us to write the RL objective as $\max_{\pi \in \Pi} \mathbb{E}_{s_1 \sim \mu}[V_1^\pi(s_1)]$, where $\Pi$ is the set of all the stationary policies. Equivalently, we can rewrite the RL objective into its dual formulation [44], i.e.,

**RL**

$$\max_{\pi \in \Pi} \ \big(r \cdot d^\pi\big) =: \mathcal{J}_\infty(\pi) \tag{1}$$

where we denote with $r \in \mathbb{R}^S$ a reward vector, and with $d^\pi$ the state distribution induced by $\pi$. We call the problem (1) the *infinite trials* RL formulation. Indeed, the objective $\mathcal{J}_\infty(\pi)$ considers the sum of the rewards collected during an episode, i.e., $r \cdot d^\pi$, that we can achieve on the average of an infinite number of episodes drawn with $\pi$. This is due to the state distribution $d^\pi$ being an expectation of empirical distributions $d^\pi = \mathbb{E}_{d \sim p^\pi}[d]$. However, in practice, we can never draw infinitely many episodes following a policy $\pi$. Instead, we draw a small batch of episodes $d_n \sim p_n^\pi$. Thus, we can instead conceive a *finite trials* RL formulation that is closer to what is optimized in practice.

---

[2] We will call a sequence of states and actions a *trajectory* or a *history* indifferently.

Table 1: Various convex RL objectives and applications. The last column states the equivalence between infinite trials and finite trials settings, as derived in Proposition 1 (Appendix).

| Objective $\mathcal{F}$ | | Application | Infinite $\equiv$ Finite |
|---|---|---|---|
| $r \cdot d$ | $r \in \mathbb{R}^S, d \in \Delta(\mathcal{S})$ | RL | ✓ |
| $\|d - d_E\|_p^p$ $\mathrm{KL}(d\|d_E)$ | $d, d_E \in \Delta(\mathcal{S})$ | Imitation Learning | ✗ |
| $-d \cdot \log(d)$ | $d \in \Delta(\mathcal{S})$ | Pure Exploration | ✗ |
| $\mathrm{CVaR}_\alpha[r \cdot d]$ $r \cdot d - \mathbb{V}\mathrm{ar}[r \cdot d]$ | $r \in \mathbb{R}^S, d \in \Delta(\mathcal{S})$ | Risk-Averse RL | ✗ |
| $r \cdot d$, s.t. $\lambda \cdot d \leq c$ | $r, \lambda \in \mathbb{R}^S, c \in \mathbb{R}, d \in \Delta(\mathcal{S})$ | Linearly Constrained RL | ✓ |
| $-\mathbb{E}_z \mathrm{KL}(d_z \| \mathbb{E}_k d_k)$ | $z \in \mathbb{R}^d, d_z, d_k \in \Delta(\mathcal{S})$ | Diverse Skill Discovery | ✗ |

---

**Finite Trials RL**

$$\max_{\pi \in \Pi} \left( \mathbb{E}_{d_n \sim p_n^\pi} \left[ r \cdot d_n \right] \right) =: \mathcal{J}_n(\pi) \tag{2}$$

One could then wonder whether optimizing the finite trials objective (2) leads to results that differ from the infinite trials one (1). To this point, it is straightforward to see that the two objective functions are actually equivalent

$$\mathcal{J}_n(\pi) = \mathbb{E}_{d_n \sim p_n^\pi} \left[ r \cdot d_n \right] = r \cdot \mathbb{E}_{d_n \sim p_n^\pi} \left[ d_n \right] = r \cdot d^\pi = \mathcal{J}_\infty(\pi),$$

since $r$ is a constant vector and the expectation is a linear operator. It follows that the infinite trials and the finite trials RL formulations admit the same optimal policies. Hence, one can enjoy both the computational tractability of the infinite trials formulation and, at the same time, optimize the objective function that is employed in practice. In the next section, we will show that a similar result does not hold true for the convex RL formulation.

## 4 Convex Reinforcement Learning in Finite Trials

Even though the RL formulation covers a wide range of sequential decision-making problems, several relevant applications cannot be expressed by means of the inner product between a linear reward vector $r$ and a state distribution $d^\pi$ [2, 49]. These include imitation learning, pure exploration, constrained problems, and risk-sensitive objectives, among others. Recently, a *convex* RL formulation [60, 57, 21] has been proposed to unify these applications in a unique general framework, which is

**Convex RL**

$$\max_{\pi \in \Pi} \left( \mathcal{F}(d^\pi) \right) =: \zeta_\infty(\pi) \tag{3}$$

where $\mathcal{F} : \Delta(S) \to \mathbb{R}$ is a function[3] of the state distribution $d^\pi$. In Table 1, we recap some of the most relevant problems that fall under the convex RL formulation, along with their specific $\mathcal{F}$ function. As it happens for linear RL, in any practical simulated or real-world scenario, we can only draw finite number of episodes with a policy $\pi$. From these episodes, we obtain an empirical distribution $d_n \sim p_n^\pi$ rather than the actual state distribution $d^\pi$, where $n$ is the number of episodes. This can cause a mismatch from the objective that is typically considered in convex RL [60, 57], and what can be optimized in practice. To overcome this mismatch, we define a finite trials formulation of the convex RL objective, as we did in the previous section for the linear RL formulation.

**Finite Trials Convex RL**

$$\max_{\pi \in \Pi} \left( \mathbb{E}_{d_n \sim p_n^\pi} \left[ \mathcal{F}(d_n) \right] \right) =: \zeta_n(\pi) \tag{4}$$

---

[3]In this context, we use the term *convex* to distinguish it from the standard *linear* RL objective. However, in the following we will consider functions $\mathcal{F}$ that are either convex, concave or even non-convex. In general, problem (3) takes the form of a max problem for concave $\mathcal{F}$, or a min problem for convex $\mathcal{F}$.

Comparing objectives (3) and (4), one can notice that both of them include an expectation over the episodes, being $d^\pi = \mathbb{E}_{d \sim p^\pi}[d]$. Especially, we can write

$$\zeta_\infty(\pi) = \mathcal{F}(d^\pi) = \mathcal{F}(\mathbb{E}_{d_n \sim p_n^\pi}[d_n]) \leq \mathbb{E}_{d_n \sim p_n^\pi}[\mathcal{F}(d_n)] = \zeta_n(\pi)$$

through the Jensen's inequality. As a consequence, optimizing the infinite trials objective $\zeta_\infty(\pi)$ does not necessarily lead to an optimal behavior for the finite trials objective $\zeta_n(\pi)$. From a mathematical perspective, this is due to the fact that the empirical distributions $d_n$ induced by the policy $\pi$ are averaged by the expectation $d^\pi$ before computing the $\mathcal{F}$ function into objective (3), thus losing a measure of the performance $\mathcal{F}$ for each batch of episodes, which we instead keep in the objective (4).

## 4.1  Approximating the Finite Trials Objective with Infinite Trials

Despite the evident mismatch between the finite trials and the infinite trials formulation of the convex RL problem, most existing works consider (3) as the standard objective, even if only a finite number of episodes can be drawn in practice. Thus, it is worth investigating how much we can lose by approximating a finite trials objective with an infinite trials one. First, we report a useful assumption on the structure of the function $\mathcal{F}$.

**Assumption 4.1** (Lipschitz). *A function $\mathcal{F} : \mathcal{X} \to \mathbb{R}$ is Lipschitz-continuous if it holds for some constant $L$*

$$\left| \mathcal{F}(x) - \mathcal{F}(y) \right| \leq L \left\| x - y \right\|_1, \qquad \forall (x, y) \in \mathcal{X}^2.$$

Then, we provide an upper bound on the approximation error in the following theorem.

**Theorem 4.1** (Approximation Error). *Let $n \in \mathbb{N}$ be a number of trials, let $\delta \in (0, 1]$ be a confidence level, let $\pi^\dagger \in \arg\max_{\pi \in \Pi} \zeta_n(\pi)$ and $\pi^\star \in \arg\max_{\pi \in \Pi} \zeta_\infty(\pi)$. Then, it holds with probability at least $1 - \delta$*

$$err := \left| \zeta_n(\pi^\dagger) - \zeta_n(\pi^\star) \right| \leq 4LT \sqrt{\frac{2S \log(4T/\delta)}{n}}$$

The previous result establishes an approximation error rate $err = O(LT\sqrt{S/n})$ that is polynomial in the number of episodes $n$. Unsurprisingly, the guarantees over the approximation error scales with $O(1/\sqrt{n})$, as one can expect the empirical distribution $d_n$ to concentrate around its expected value for large $n$ [54]. This implies that approximating the finite trials objective $\zeta_n(\pi)$ with the infinite trials $\zeta_\infty(\pi)$ can be particularly harmful in those settings in which $n$ is necessarily small. As an example, consider training a robot through a simulator and deploying the obtained policy in the real world, where the performance measures are often based on a single episode ($n = 1$). The performance that we experience from the deployment can be much lower than the expected $\zeta_\infty(\pi)$, which might result in undesirable or unsafe behaviors. However, Theorem 4.1 only reports an instance-agnostic upper bound, and it does not necessarily imply that there would be a significant approximation error in a specific instance, i.e., a specific pairing of an MDP $\mathcal{M}$ and a function $\mathcal{F}$. Nevertheless, in this paper we argue that the upper bound of the approximation error is not vacuous in several relevant applications, and we provide an illustrative numerical corroboration of this claim in Section 6.

**Challenged assumption 1.** *The convex RL problem can be equivalently addressed with an infinite trials formulation.*

Finally, in Figure 2 we report a visual representation[4] of the approximation error defined in Theorem 4.1. Notice that the finite trials objective $\zeta_n$ converges uniformly to the infinite trials objective $\zeta_\infty$ as a trivial consequence of Theorem 4.1. This is particularly interesting as it results in $\pi^\dagger$ converging to $\pi^\star$ in the limit of large $n$ as shown Figure 2.

# 5  In-Depth Analysis of Single Trial Convex Reinforcement Learning Setting

Having established a significant mismatch between the infinite trials convex RL setting that is usually considered in previous works, i.e., $\zeta_\infty(\pi)$, and the finite trials formulation that is actually targeted in

---

[4]Note that it is not possible to represent the objective functions in two dimensions in general. Nevertheless, we provide an abstract one-dimensional representation of the policy space to bring the intuition.

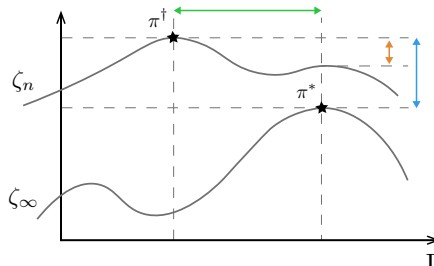 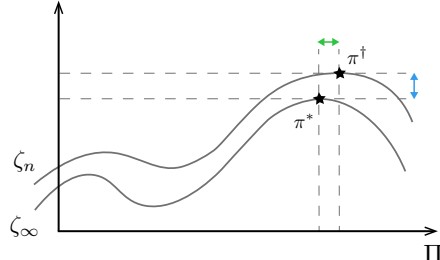

Figure 2: The two illustrations report an abstract visualization of $\zeta_n$ and $\zeta_\infty$ for small values of $n$ (left) and large values of $n$ (right) respectively. The green bar visualize the distance $\left\| d_n - d^{\pi^\star} \right\|_1$, in which $d_n \sim p_n^{\pi^\dagger}$. The blue bar visualize the distance $\left| \zeta_n(\pi^\dagger) - \zeta_\infty(\pi^\star) \right|$. The orange bar visualize the approximation error, i.e., the distance $\left| \zeta_n(\pi^\dagger) - \zeta_n(\pi^\star) \right|$.

practice, i.e., $\zeta_n(\pi)$, it is now worth taking a closer look at the finite trials optimization problem (4). Indeed, to avoid the approximation error that can occur by optimizing (4) through the infinite trials formulation (Theorem 4.1), one could instead directly address the optimization of (4). Especially, how does the finite trials convex RL problem compare to its infinite trials formulation and the linear RL problem? What kind of policies do we need to optimize the finite trials objective? Is the underlying learning process statistically harder than infinite trials convex RL? In this section, we investigate the answers to these relevant questions. To this purpose, we will focus on a *single trial* setting, i.e., $\zeta_n(\pi)$ with $n = 1$, which allows for a clearer analysis, while analogous considerations should extend to a general number of trials $n > 1$.

**Single Trial Convex RL**

$$\max_{\pi \in \Pi} \left( \mathop{\mathbb{E}}_{d \sim p^\pi} \left[ \mathcal{F}(d) \right] \right) =: \zeta_1(\pi) \tag{5}$$

Taking inspiration from [57], we can cast the problem (5) defined over an MDP $\mathcal{M}$ into a *convex* MDP $\mathcal{CM} := (\mathcal{S}, \mathcal{A}, P, T, \mu, \mathcal{F})$, where $\mathcal{S}, \mathcal{A}, P, T, \mu$ are defined as in a standard MDP (see Section 2), and $\mathcal{F} : \Delta(\mathcal{S}) \to \mathbb{R}$ is a convex function that defines the objective $\zeta_1(\pi)$. Is solving a convex MDP $\mathcal{CM}$ significantly harder than solving an MDP $\mathcal{M}$?

### 5.1 Extended MDP Formulation of the Single Trial Convex RL Setting

We can show that any finite-horizon convex MDP $\mathcal{CM}$ can be actually translated into an equivalent MDP $\mathcal{M}_\ell = (\mathcal{S}_\ell, \mathcal{A}_\ell, P_\ell, \mu_\ell, r_\ell)$, which we call an *extended* MDP. The main idea is to temporally-extend $\mathcal{CM}$ so that each state contains the information of the full trajectory leading to it, so that the convex objective can be cast into a linear reward. To do this, we define the extended state space $\mathcal{S}_\ell$ to be the set of all the possible histories up to length $T$, so that $s_\ell \in \mathcal{S}_\ell$ now represents a history. Then, we can keep $\mathcal{A}_\ell, P_\ell, \mu_\ell$ equivalent to $\mathcal{A}, P, \mu$ of the original $\mathcal{CM}$, where for the extended transition model $P_\ell(s'_\ell | s_\ell, a)$ we solely consider the last state in the history $s_\ell$ to define the conditional probability to the next history $s'_\ell$. Finally, we just need to define a scalar reward function $r_\ell : \mathcal{S}_\ell \to \mathbb{R}$ such that $r_\ell(s_\ell) = \mathcal{F}(d_{s_\ell})$ for all the histories $s_\ell$ of length $T$ and $r_\ell(s_\ell) = 0$ otherwise, where we denoted with $d_{s_\ell}$ the empirical state distribution induced by $s_\ell$.

Notably, the problem of finding an optimal policy for the extended MDP $\mathcal{M}_\ell$, i.e., $\pi^* \in \arg\max_{\pi \in \Pi} r_\ell \cdot d^\pi$, is equivalent to solve the problem (5). Indeed, we have

$$r_\ell \cdot d^\pi = \sum_{s_\ell \in \mathcal{S}_\ell} r_\ell(s_\ell) d^\pi(s_\ell) = \sum_{s_\ell \in \mathcal{S}_\ell} \mathcal{F}(d_{s_\ell}) \mathbb{1}(|s_\ell| = T) p^\pi(d_{s_\ell}) = \mathop{\mathbb{E}}_{d_{s_\ell} \sim p^\pi} [\mathcal{F}(d_{s_\ell})].$$

Whereas $\mathcal{M}_\ell$ can be solved with classical MDP methods [44], the size of the policy $\pi : \mathcal{S}_\ell \to \Delta(\mathcal{A}_\ell)$ to be learned does scale with the size of $\mathcal{M}_\ell$, which grows exponentially in the episode horizon as we have $|\mathcal{S}_\ell| > S^T$. Thus, the extended MDP formulation and the resulting insight cast some doubts on the notion that the convex RL is not significantly harder than standard RL [60, 57].

**Challenged assumption 2.** *Convex RL is only slightly harder than the standard RL formulation.*

## 5.2 Partially Observable MDP Formulation of the Single Trial Convex RL Setting

Instead of temporally extending the convex MDP $\mathcal{CM}$ as in the previous section, which causes the policy space to grow exponentially with the episode horizon $T$, we can alternatively formulate $\mathcal{CM}$ as a Partially Observable MDP (POMDP) [5, 31] $\mathcal{PM} = (\mathcal{S}_\ell, \mathcal{A}_\ell, P_\ell, \mu_\ell, r_\ell, \Omega, O)$, in which $\Omega$ denotes an observation space, and $O : \mathcal{S}_\ell \to \Delta(\Omega)$ is an observation function. The process to build $\mathcal{PM}$ is rather similar to the one we employed for the extended MDP $\mathcal{M}_\ell$, and the components $\mathcal{S}_\ell, \mathcal{A}_\ell, P_\ell, \mu_\ell, r_\ell$ remain indeed unchanged. However, in a POMDP the agent does not directly access a state $s_\ell \in \mathcal{S}_\ell$, but just a partial observation $o \in \Omega$ that is given by the observation function $O$. Here $O(s_\ell) = o$ is a deterministic function such that the given observation $o$ is the last state in the history $s_\ell$. Since the agent only observes $o$, a stationary policy can be defined as a function $\pi : \Omega \to \Delta(\mathcal{A})$, for which the size depends on the number of states $S$ of the convex MDP $\mathcal{CM}$, being $\Omega = \mathcal{S}$. However, it is well known [31] that history-dependent policies should be considered for the problem of optimizing a POMDP. This is in sharp contrast with the current convex MDP literature, which only considers stationary policies due to the infinite trials formulation [60].

**Challenged assumption 3.** *The set of stationary randomized policies is sufficient for convex RL.*

## 5.3 Online Learning in Single Trial Convex RL

Let us assume to have access to a planning oracle that returns an optimal policy $\pi^*$ for a given $\mathcal{CM}$, so that we can sidestep the concerns on the computational feasibility reported in previous sections. It is worth investigating the complexity of learning $\pi^*$ from *online interactions* with an unknown $\mathcal{CM}$. A typical measure of this complexity is the online learning regret $\mathcal{R}(N)$, which is defined as

$$\mathcal{R}(N) := \sum_{t=1}^{N} V^* - V^{(t)},$$

where $N$ is the number of learning episodes, $V^* = V_1^{\pi^*}(s_1)$ is the value of the optimal policy, $V^{(t)} = V^{\pi_t}$ is the value of the policy $\pi_t$ deployed at the episode $t$. We now aim to assess whether there exists a principled algorithm that can achieve a sub-linear regret $\mathcal{R}(N)$ in the worst case. To this purpose, we can cast our learning problem in the Once-Per-Episode (OPE) RL formulation [12]. In the latter setting, the agent interacts with the MDP for $T$ steps, receiving a $0/1$ feedback at the end of the episode, where the feedback is computed according to a logistic model that is function of the history. To translate our objective $\zeta_1(\pi) = \mathbb{E}_{d \sim p^\pi}[\mathcal{F}(d)]$ into the OPE framework [12], we have to encode $\mathcal{F}$ into a linear representation. With the following assumption, we state the existence of such representation.

**Assumption 5.1** (Linear Realizability). *The function $\mathcal{F}$ is linearly-realizable if it holds*

$$\mathcal{F}(d) = \mathbf{w}_*^\top \phi(d),$$

*where $\mathbf{w}_* \in \mathbb{R}^{d_\mathbf{w}}$ is a vector of parameters such that $\|\mathbf{w}_*\|_2 \leq B$ for some known $B > 0$, and $\phi(d) = (\phi_j(d))_{j=1}^{d_\mathbf{w}}$ is a known vector of basis functions such that $\|\phi(d)\|_2 \leq 1, \forall d \in \Delta(\mathcal{S})$.*

With the Assumption 5.1 and other minor changes that are detailed in the Appendix, we can invoke the analysis of OPE-UCBVI in [12] to provide an upper bound to the regret $\mathcal{R}(N)$ in our setting.

**Theorem 5.1** (Regret). *For any confidence $\delta \in (0, 1]$ and unknown convex MDP $\mathcal{CM}$, the regret of the* OPE-UCBVI *algorithm is upper bounded as*

$$\mathcal{R}(N) \leq O\left(\left[d_\mathbf{w}^{7/2} B^{3/2} T^2 S A^{1/2}\right]\sqrt{N}\right)$$

*with probability $1 - \delta$.*

The latter result states that the problem of learning an optimal policy in a unknown convex MDP is at least statistically efficient assuming linear realizability and the access to a planning oracle. Those are fairly strong assumptions, but principled approximate solvers may be designed to overcome the planning oracle assumption, whereas in several convex RL settings the function $\mathcal{F}$ is assumed to be known, and thus trivially realizable.

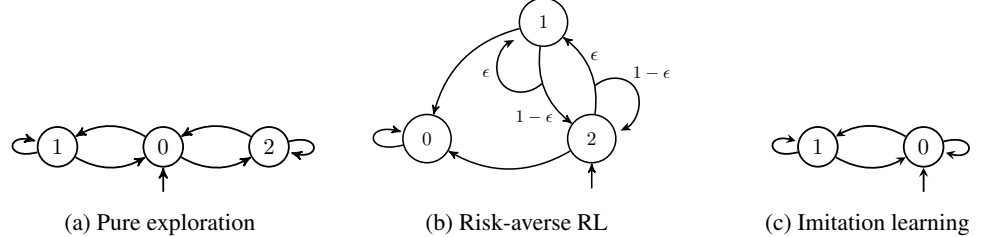

|                     |                      |                        |
|:-------------------:|:--------------------:|:----------------------:|
| (a) Pure exploration | (b) Risk-averse RL  | (c) Imitation learning |

Figure 3: Visualization of the illustrative MDPs. In **(b)**, state $0$ is a low-reward ($r$) low-risk state, state $2$ is a high-reward ($R$) high-risk state, and state $1$ is a penalty state with zero reward.

## 6 Numerical Validation

In this section, we evaluate the performance over the finite trials objective (4) achieved by a policy $\pi^{\dagger} \in \arg\max_{\pi \in \Pi} \zeta_n(\pi)$ maximizing the same finite trials objective (4) against a policy $\pi^{\star} \in \arg\max_{\pi \in \Pi} \zeta_{\infty}(\pi)$ maximizing the infinite trials objective (3) instead. The latter infinite trials $\pi^{*}$ can be obtained by solving a dual optimization on the convex MDP (see Sec. 6.2 in [41]),

$$\max_{\omega \in \Delta(\mathcal{S} \times \mathcal{A})} \mathcal{F}(\omega), \qquad \text{subject to} \sum_{a \in \mathcal{A}} \omega(s, a) = \sum_{s' \in \mathcal{S}, a' \in \mathcal{A}} P(s|s', a')\omega(s', a'), \ \forall s \in \mathcal{S},$$

To get the finite trials $\pi^{\dagger}$, we first recover the extended MDP as explained in Section 5.1, and then we apply standard dynamic programming [7]. In the experiments, we show that optimizing the infinite trials objective can lead to sub-optimal policies across a wide range of applications. In particular, we cover examples from pure exploration, risk-averse RL, and imitation learning. We carefully selected MDPs that are as simple as possible in order to stress the generality of our results. For the sake of clarity, we restrict the discussion to the single trial setting ($n = 1$).

**Pure Exploration**  For the pure exploration setting, we consider the state entropy objective [27], i.e., $\mathcal{F}(d) = H(d) = -d \cdot \log d$, and the convex MDP in Figure 3a. In this example, the agent aims to maximize the state entropy over finite trajectories of $T$ steps. Notice that this happens when a policy induces an empirical state distribution that is close to uniform. In Figure 4a, we compare the average entropy induced by the optimal finite trials policy $\pi^{\dagger}$ and the optimal infinite trials policy $\pi^{\star}$. An agent following the policy $\pi^{\dagger}$ always achieves a uniform empirical state distribution leading to the maximum entropy. Moreover, $\pi^{\dagger}$ is a non-Markovian deterministic policy. In contrast, the policy $\pi^{*}$ is randomized in all the three states. As a result, this policy induces sub-optimal empirical state distributions with *strictly positive* probability, as shown in Figure 4d.

**Risk-Averse RL**  For the risk-averse RL setting, we consider a Conditional Value-at-Risk (CVaR) objective [46] given by $\mathcal{F}(d) = \text{CVaR}_{\alpha}[r \cdot d]$, where $r \in [0, 1]^{S}$ is a reward vector, and the convex MDP in Figure 3b, in which the agent aims to maximize the CVaR over a finite-length trajectory of $T$ steps. First, notice that a financial semantics can be attributed to the given MDP. An agent, starting in state 2, can decide whether to invest in risky assets, e.g., crypto-currencies, or in safe ones, e.g., treasury bills. Because of the stochastic transitions, a policy would need to be reactive to the realizations of the transition model in order to maximize the single trial objective (5). This kind of behavior is achieved by an optimal finite trials policy $\pi^{\dagger}$. Indeed, $\pi^{\dagger}$ is a non-Markovian deterministic policy, which can take decisions as a function of history, and thus takes into account the current realization. On the other hand, an optimal infinite trials policy $\pi^{*}$ is a Markovian policy, and it cannot take into account the current history. As a result, the policy $\pi^{*}$ induces sub-optimal trajectories with *strictly positive* probability (see Figure 4e). Finally, in Figure 4b we compare the single trial performance induced by the optimal single trial policy $\pi^{\dagger}$ and the optimal infinite trials policy $\pi^{\star}$. Overall, $\pi^{\dagger}$ performs significantly better than $\pi^{\star}$.

**Imitation Learning**  For the imitation learning setting, we consider the distribution matching objective [32], i.e., $\mathcal{F}(d) = \text{KL}\,(d\|d_E)$, and the convex MDP in Figure 3c. The agent aims to learn a policy $\pi$ inducing an empirical state distribution $d$ close to the empirical state distribution $d_E$ demonstrated by an expert. In Figure 4c, we compare single trial performance induced by the optimal single trial policy $\pi^{\dagger}$ and the optimal infinite trials policy $\pi^{\star}$. An agent following $\pi^{\dagger}$ induces an empirical state distribution that perfectly matches the expert. In contrast, an agent following $\pi^{*}$ induces sub-optimal realizations with *strictly positive* probability (see Figure 4f).

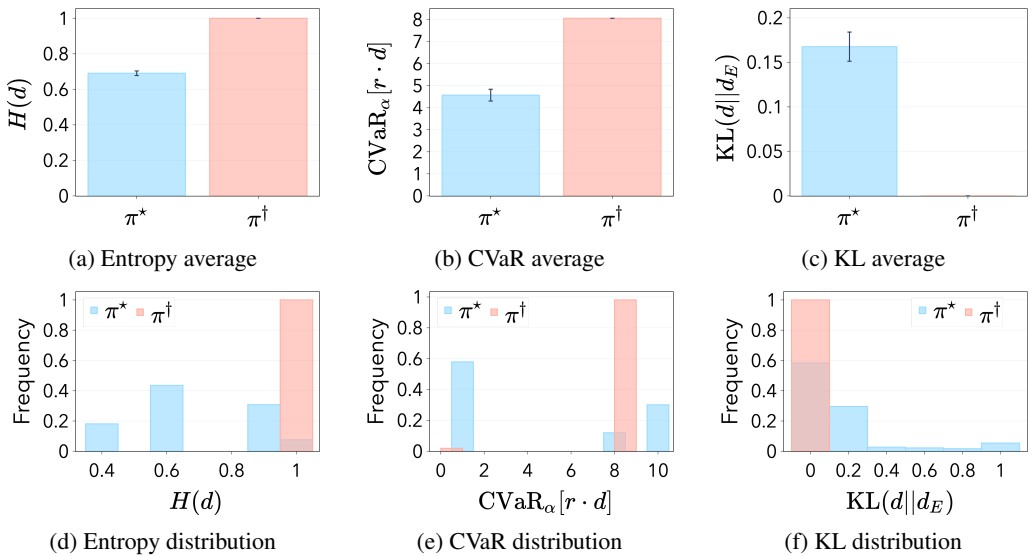

Figure 4: $\pi^{\dagger}$ denotes an optimal single trial policy, $\pi^{\star}$ denotes an optimal infinite trials policy. In **(a, d)** we report the average and the empirical distribution of the single trial utility $H(d)$ achieved in the pure exploration convex MDP ($T = 6$) of Figure 3a. In **(b, e)** we report the average and the empirical distribution of the single trial utility $\mathrm{CVaR}_\alpha[r \cdot d]$ (with $\alpha = 0.4$) achieved in the risk-averse convex MDP ($T = 5$) of Figure 3b. In **(c, f)** we report the average and the empirical distribution of the single trial utility $\mathrm{KL}(d||d_E)$ (with expert distribution $d_E = (1/3, 2/3)$) achieved in the imitation learning convex MDP ($T = 12$) of Figure 3c. For all the results, we provide 95 % c.i. over 1000 runs.

## 7 Related Work

To the best of our knowledge, [27] were the first to introduce the convex RL problem, as a generalization of the standard RL formulation to non-linear utilities, especially the entropy of the state distribution. They show that the convex RL objective, while being concave (convex) in the state distribution, can be non-concave (non-convex) in the policy parameters. Anyway, they provide a provably efficient algorithm that overcomes the non-convexity through a Frank-Wolfe approach. [60] study the convex RL problem under the name of RL with general utilities. Especially, they investigated a hidden convexity of the convex RL objective that allows for statistically efficient policy optimization in the infinite-trials setting. Recently, the infinite trials convex RL formulation has been reinterpreted from game-theoretic perspectives [57, 21]. The former [57] notes that the convex RL problem can be seen as a min-max game between the policy player and a cost player. The latter [21] shows that the convex RL problem is a subclass of mean-field games.

Another relevant branch of literature is the one investigating the expressivity of scalar (Markovian) rewards [2, 49]. Especially, [2] shows that not all the notions of task, such as inducing a set of admissible policies, a (partial) policy ordering, a trajectory ordering, can be naturally encoded with a scalar reward function. Whereas the convex RL formulation extends the expressivity of scalar RL w.r.t. all these three notions of task, it is still not sufficient to cover any instance. Convex RL is powerful in terms of the policy ordering it can induce, but it is inherently limited on the trajectory ordering as it only accounts for the infinite trials state distribution. Instead, the finite trials convex RL setting that we presented in this paper is naturally expressive in terms of trajectory orderings, at the expense of a diminished expressivity on the policy orderings w.r.t. infinite trials convex RL.

Previous works concerning RL in the presence of trajectory feedback are also related to this work. Most of this literature assumes an underlying scalar reward model [e.g., 18] which only delays the feedback at the end of the episode. One notable exception is the once-per-episode formulation in [12], which we have already commented on in Section 5.

Finally, the work in [13, 14] considers infinite-horizon MDPs with vectorial rewards as a mean to encode convex objectives in RL with a multi-objective flavor. They show that stationary policies are in general sub-optimal for the introduced online learning setting, where non-stationarity becomes

essential. In this setting, they provide principled procedures to learn an optimal policy with sub-linear regret. Their work essentially complement our analysis in the infinite-horizon problem formulation, where the difference between finite trials and infinite trials fades away.

## 8 Conclusion and Future Directions

While in classic RL the optimization of an infinite trials objective leads to the optimal policy for the finite trials counterpart, we have shown that true convex RL does not have this property. First, we have formalized the concept of finite trials convex RL, which captures a problem that until now has been cast into an unfounded optimization problem. Then, we have given an upper bound on the approximation error obtained by erroneously optimizing the infinite trials objective, as it is currently done in practice. Finally, we have presented intuitive, yet general, experimental examples to show that the approximation error can be significant in relevant applications. Since the finite trials setting is the standard in both simulated and real-world RL, we believe that shedding light on the above mentioned performance gap will lead to better approaches for convex RL and related areas. Future work could target approximate solutions to the finite trials objective rather than the infinite trials one, which can cause sub-optimality even when solved exactly. Methods in POMDPs [26, 34] or optimistic planning algorithms [10] could provide useful inspiration.

## Acknowledgements

Riccardo De Santi and Piersilvio De Bartolomeis thank professor Niao He for offering graduate students at ETH the opportunity to work in exciting research areas within the "Foundations of Reinforcement Learning" course. Further, we thank Ali Batuhan Yardim for his generous feedback on an early version of this work.

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
