# OpenReview forum: "Challenging Common Assumptions in Convex Reinforcement Learning"
_NeurIPS.cc/2022/Conference — NeurIPS 2022 Accept_

### Official Review · Reviewer_CnKV · 2022-07-09

**Rating:** 4
**Confidence:** 3
**Soundness:** 2 fair
**Presentation:** 2 fair
**Contribution:** 2 fair

**Summary:**

This paper shows that optimizing the objective posed by "convex RL" using a finite number of trials can lead to significant differences in policy performance when compared to what the analysis yields under infinite trails. They argue that this approximation error can be decreased by explicitly accounting for the use of a finite number of trials in the convex RL objective, and verify this gap in performance both theoretically and empirically.

**Questions:**

## Some comments:
Abstract:
1. "More recently, convex RL has been introduced to extend the RL formulation to all the objectives that are convex functions of the state distribution induced by a policy." -- I would rephrase this to address the fact that recent work can provide solutions in non-stationary MDPs. What the reward is a function of is only important in so much as it changes with the policy distribution.
2. "Notably, convex RL covers several relevant applications that do not fall into the scalar formulation, including imitation learning, risk-averse RL, and pure exploration" --This statement is confusing are you proposing to optimize vector valued rewards or rewards that are convex in the induced state distribution of the policy.

Introduction:
1. "The coefficients of this linear combination are given by the state visitation distribution induced by the agent’s policy." -- This is not true for general RL just the variant which is discussed in this paper and RL under general utilities paper.
2. "Thus, the objective function can be equivalently written as the inner product between the mentioned state distribution and a reward vector." -- Again not true for general RL, only the tabular settings in which there are a finite number of states and actions.
3. "Several works have thus extended the standard RL formulation to address non-linear objectives of practical interest." -- I would not call imitation learning an extension of RL. In most cases its a much simpler problem, unless you are actually talking about apprenticeship learning or imitation learning from observation alone (e.g. only states induced by an expert policy are observed).
4. "All this large body of work has been recently unified into a unique framework, called convex RL" --Please just explicitly state what the assumptions for convex RL are, then explain why each of these sub-fields falls into this framework.
5. "The convex RL problem has been showed to be largely tractable either computationally, as it admits a dual formulation akin to standard RL [24], or statistically, as principled algorithms achieving sub-linear regret rates that are slightly worse than standard RL have been developed [31, 30]." --This sentence is poorly worded, and makes it hard for the reader to understand what is being asserted.
6. "This has never been a problem in standard RL: due to the scalar objective, optimizing the policy over infinite trials or finite trials is equivalent, as it leads to the same optimal policy" -- Please be more clear, the use of equivalent is vague and it does not leave the reader with a notion of what is lost when we don't work with exact expectations. Also the assertion that using finite numbers of samples does not affect standard RL problems is definitely not true in general. Please make this statement more precise.
7. "a policy optimized over infinite trials can be significantly sub-optimal when deployed over finite trials" -- Did you mean to say, '`'significantly more sub-optimal than a policy  optimized over finite trials'?
8. "In the real world, we can only deploy the strategy over a single trial. Thus, we are only interested in the performance of the strategy in the real-world realization, rather  than the performance of the strategy when averaging different realization" -- Again this phrasing really throws me off. Just say that "in the real world we gather finite numbers of examples so we should consider algorithms which explicitly use a finite number of examples during training" if that is indeed what you are trying to say.
9. "(1) convex RL can be equivalently addressed with an infinite trials formulation" -- This is extremely vague. Please make your assertion more direct and testable. Especially when this is the core motivation for the paper.
10. "  (3) stationary policies are in general sufficient" This also needs to be made more clear. You have not introduced a notion of stationarity, nor have you explicitly stated for what they are sufficient. Without this clarity inclusion of this statement in the introduction is meaningless.

Background
1. Making the reward a function only of state is non-standard.
2. You have multiple definitions of \pi_t that are inconsistent with one-another. In fact the whole destinction between stationary and non-stationary is unclear. What do you mean by a time-consistent rule? In the classic RL literature, if I am not mistaken, stationary usually means that the policy produces a value function which satisfies the bellman equations.
3. Please explicitly state your assumptions about the class of MDPs you consider, namely tabular.

Reinforcement learning with Finite trails
1. Again much of your computational tractability comes from the assumption on the reward which is not true for general RL.

Convex Reinforcement Learning in Finite Trials
1. All of this is predicated on the idea that the expectation is taken before the model is run. Which in general its not, and many standard RL algorithms function and have been analyzed in this regime.
2. There is formatting issue with 4.1
3. I am uncertain about the claim from Theorem 4.1. Are you proposing this as a this a novel contribution of this paper? I know that even extremely old Apprenticeship learning papers analyze the solution dependency between exact and inexact expectations, namely (1) and (2). There has also been more work following mirror-decent algorithms that even inexact updates can provide algorithms with sub-linear regret (3). I am not sure if these extend to "Convex RL" with general utilities but I am not sure why they would not. If they don't please comment on the limitations of online mirror decent algorithms.
3. "Challenged assumption 1. The convex RL problem can be equivalently addressed with an infinite trials formulation" -- What does equivalently addressed mean? If this is one of the main points of the paper it needs to be a precise statement.

In-Depth Analysis of Single Trial Convex Reinforcement Learning Setting
1. "Thus, the extended MDP formulation and the resulting insight cast some doubts on the notion that the convex RL is not significantly harder than standard RL [31, 30]." -- Just state what the finding is. Your statement here is so conservative so as to be basically meaningless.
2. Its extremely difficult to untangle what exactly is being stated by 5.2.
3. "The set of stationary randomized policies is sufficient for convex RL." --Sufficient for what.
4. Formatting issues with Theorem 5.1
5. Are you claiming that Theorem 5.1 is unique in providing sub-linear regret for the task of RL under general utilities in the presence of function approximation / inexact optimization / inexact expectations? It seems as though there must be result that are similar (e.g. provide sub-linear regret in the online mirror decent literature. Might look here (3)).

Illustrative Examples and Experimental Approximation Error
1. These are nice toy examples but if the point of this paper was to provide a better bridge between theory and practice please include an example which actually has a reasonably large state action space. Most problems that are used in practice e.g. autonomous vehicles or estimation of volatility usually have horizons and state-action spaces that are significantly larger then what is described here. Without a stronger, or clearer theoretical contribution I think more experiments illustrating your point would be helpful.

# Citations Mentioned
(1) https://dl.acm.org/doi/abs/10.1145/1015330.1015430

(2) https://proceedings.neurips.cc/paper/2007/file/ca3ec598002d2e7662e2ef4bdd58278b-Paper.pdf

(3) https://arxiv.org/pdf/2102.06924.pdf

**Limitations:**

1. Poor experimental evaluation.
2. Incremental theoretical contribution.
3. Poor writing quality and paper organization.
4. Code not included in supplemental material.

**Strengths And Weaknesses:**

# Strengths
1. I really like the intended theme of the paper. I think that many results that I have seen in terms of work in online learning and minimax problems ignore crucial issues that lead to either algorithms that function poorly in practice or bounds that are effectively meaningless.
2. I think focusing on different assumptions that don't hold, or produce vacuous bounds is a great way to move the field forward.

# Weaknesses
1. Almost all the challenged assumption statements are imprecise and difficult to interpret.
2. All experiments  are on examples that are so small as to be unrealistic for problem classes which are considered in practice.
3. The statement of assumptions is extremely sparse, making verification of the novelty of this work difficult.
4. The statement of the results and how to interpret them was also quite confusing.
5. Work ignores some of the first apprenticeship learning algorithms that actively analyze how working with inexact state-occupancy affects algorithm performance and regret.

---

> ### Author Response · Authors · 2022-08-01
> **Authors Response 1/4**
>
> We thank the Reviewer for their detailed comments. We will take into serious consideration their concerns about the wording of some sentences in the revision of the paper.
>
> We believe that the Reviewer's negative evaluation could be due to some misinterpretation of our work. Thus, we provide below clarifications on some crucial points that we see as potential sources of confusion. Then, we report detailed answers to the Reviewer's comments.
>
> We hope that these clarifications can make the Reviewer better appreciate our work and reward it for the significance of the contribution that they also recognized. If the Reviewer thinks anything has remained unanswered despite our best effort, we encourage them to ask for further clarification.
>
> **Main Clarifications**
>
> RL FORMULATION IS NOT GENERAL
>
> > "The coefficients of this linear combination are given by the state visitation distribution induced by the agent’s policy." -- This is not true for general RL just the variant which is discussed in this paper and RL under general utilities paper.
>
> > "Thus, the objective function can be equivalently written as the inner product between the mentioned state distribution and a reward vector." -- Again not true for general RL, only the tabular settings in which there are a finite number of states and actions.
>
> > Making the reward a function only of state is non-standard.
>
> > Again much of your computational tractability comes from the assumption on the reward which is not true for general RL.
>
> We are not sure whether the Reviewer is concerned by the fact that we consider state distributions instead of state-action distributions or that we consider tabular settings instead of continuous settings. Thus, we will provide answers to both concerns.
>
> From traditional MDP results (Puterman, 2014), we know that the standard RL objective function can be written as the inner product between a reward vector and the state-action distribution induced by the agent's policy (this is called the dual formulation of the MDP). In this paper, we instead consider a reward function defined over the states of the environment rather than state-actions, as it is most common in the convex RL framework [e.g., 30, 31]. However, all of the presented results are straightforward to generalize to state-action reward functions, for which we obtain similar implications. Hence, we believe that formulating the RL objective as the inner product between a reward vector and the state distribution is general.
>
> As the Reviewer noted, we only consider tabular settings in this paper. We will make this explicit in an updated version. However, we disagree that the tabular setting is not general: The tabular representation is the most general RL formulation, which admits all other settings as special cases. Our analysis indeed assumes finite states and actions, as it is typical to establish results in discrete settings before looking into function approximation. Nonetheless, we believe that most of our results would generalize to function approximation as well, as we show that the objective functions one is trying to approximate are actually different in the infinite trials and finite trials settings. Anyway, this conclusion requires further analysis that could be carried out in future work.
>
> CONVEX RL AND ASSUMPTIONS
>
> > "All this large body of work has been recently unified into a unique framework, called convex RL" --Please just explicitly state what the assumptions for convex RL are, then explain why each of these sub-fields falls into this framework.
>
> We are not sure what kind of assumptions the Reviewer is referring to. Convex RL models a learning task over an MDP where the objective is a convex/concave function of the state distributions induced by the agent. It is a generalization of RL and it works under the same assumptions. Note that we are not the first to propose this framework, which has been presented in [31, 30, 12] before. The novelty of our work is to recognize the mismatch between the infinite trials formulation (previous works) and the finite trials formulation (formalized in this work) of the convex RL problem. As for how the sub-fields fall in the convex RL formulation, the Reviewer may look at Table 1 in the paper, where we report the $\mathcal{F}$ functions that give the specific sub-fields.

---

> > ### Author Response · Authors · 2022-08-01
> > **Author response 2/4**
> >
> > MEANING OF EQUIVALENCE AND SUFFICIENCY
> >
> > > "This has never been a problem in standard RL: due to the scalar objective, optimizing the policy over infinite trials or finite trials is equivalent, as it leads to the same optimal policy" -- Please be more clear, the use of equivalent is vague and it does not leave the reader with a notion of what is lost when we don't work with exact expectations.
> >
> > > "(1) convex RL can be equivalently addressed with an infinite trials formulation" -- This is extremely vague. Please make your assertion more direct and testable. Especially when this is the core motivation for the paper.
> >
> > > " (3) stationary policies are in general sufficient" This also needs to be made more clear. You have not introduced a notion of stationarity, nor have you explicitly stated for what they are sufficient. Without this clarity inclusion of this statement in the introduction is meaningless.
> >
> > > "Challenged assumption 1. The convex RL problem can be equivalently addressed with an infinite trials formulation" -- What does equivalently addressed mean? If this is one of the main points of the paper it needs to be a precise statement.
> >
> > > "The set of stationary randomized policies is sufficient for convex RL." --Sufficient for what.
> >
> > We reported above a set of quotes in which the Reviewer is questioning the meaning of equivalence and sufficiency, for which we would like to provide clarifications.
> >
> > By *equivalence* of two problems, we mean that they admit the same objective function, and thus the same optimal solution(s). In standard RL, the infinite trials formulation and the finite trials formulation have the same objective function. One might suffer from an estimation error when doing RL in finite trials, but the estimand is exactly the same objective of the infinite trials formulation (see derivation at lines 127-128). This also means that the optimal policy for infinite trials RL is an optimal policy for finite trials RL (and vice versa). Instead, convex RL in infinite trials and finite trials have two different objective functions, and an optimal policy for the infinite trials problem is not necessarily optimal for the finite trials problem. We show throughout the paper that this mismatch is significant (Th. 4.1 and experimental validation).
> >
> > By *sufficiency* of a class of policies for a learning problem, we mean that the optimal solution for the learning problem can be found within the class of policies. E.g., saying that the class of stationary policies is sufficient for the infinite trials convex RL problem means that a stationary policy attains the global optimum of the objective function, and considering a more complex policy class (such as non-stationary policies, or history-based policies) does not yield any improvement in the optimal solution.
> >
> > PRIOR WORKS IN APPRENTICESHIP LEARNING AND NOVELTY OF OUR RESULTS
> >
> > > Work ignores some of the first apprenticeship learning algorithms that actively analyze how working with inexact state-occupancy affects algorithm performance and regret.
> >
> > > I am uncertain about the claim from Theorem 4.1. Are you proposing this as a this a novel contribution of this paper? I know that even extremely old Apprenticeship learning papers analyze the solution dependency between exact and inexact expectations, namely (1) and (2).
> >
> > We thank the Reviewer for mentioning these interesting works, but we would like to underline what makes our work crucially different.
> >
> > First, the works (1, 2) deal with an RL problem in which the (linear) reward has to be inferred from some expert’s demonstrations (through inverse RL). Thus, they are not an instance of convex RL, but they fall in the traditional RL formulation. We look at problems where a (linear) reward function does not exist, and the objective function can be instead expressed as a convex/concave function of the state distribution.
> >
> > Second, their theoretical analysis concerns the estimation error over the objective function due to the limited number of samples. Instead, we study the mismatch between the infinite trials and the finite trials convex RL objective functions, which is non-zero even when the two functions are estimated exactly.
> >
> > We hope that these clarifications can convince the Reviewer that the novelty of our results is not affected by these prior works, as they provide answers to different questions.

---

> > > ### Author Response · Authors · 2022-08-01
> > > **Authors Response 3/4**
> > >
> > > **Detailed Comments**
> > >
> > > > "Notably, convex RL covers several relevant applications that do not fall into the scalar formulation, including imitation learning, risk-averse RL, and pure exploration" --This statement is confusing are you proposing to optimize vector valued rewards or rewards that are convex in the induced state distribution of the policy.
> > >
> > > With scalar formulation, we mean the traditional RL formulation in this sentence, in which the objective is a linear function of the state distribution. We will substitute “scalar” with “linear” to avoid this confusion. The setting we care about is the convex RL setting, where the objective is a convex/concave function of the state distribution.
> > >
> > > > I would not call imitation learning an extension of RL. In most cases its a much simpler problem, unless you are actually talking about apprenticeship learning or imitation learning from observation alone
> > >
> > > The imitation learning formulation we are referring to is the one reported in Table 1, where the objective function is a distance/divergence between the state distributions induced by the agent and the expert.
> > >
> > > > Also the assertion that using finite numbers of samples does not affect standard RL problems is definitely not true in general.
> > >
> > > We are not claiming that having a finite number of samples does not affect standard RL. We claim that the objective function remains the same in both the infinite and finite trials formulation. One can still suffer from poor estimation with a limited number of samples, but the estimation target is the same objective.
> > >
> > > > "The convex RL problem has been showed to be largely tractable either computationally, as it admits a dual formulation akin to standard RL [24], or statistically, as principled algorithms achieving sub-linear regret rates that are slightly worse than standard RL have been developed [31, 30]." -- This sentence is poorly worded, and makes it hard for the reader to understand what is being asserted.
> > >
> > > Here we mean that the infinite trials convex RL formulation is computationally tractable, i.e., there exists an algorithm that computes the optimal policy in a polynomial number of steps when the MDP is known, and statistically tractable, i.e., there exists an algorithm that learns a slightly sub-optimal policy by taking a polynomial number of samples.
> > >
> > > > "a policy optimized over infinite trials can be significantly sub-optimal when deployed over finite trials" -- Did you mean to say, ''significantly more sub-optimal than a policy optimized over finite trials’’?
> > >
> > > We mean that the optimal policy for the infinite trials objective can be sub-optimal for the finite trials objective, which admits another optimal policy in general. The gap between the two policies can be significant, as shown in Th. 4.1.
> > >
> > > > "(1) convex RL can be equivalently addressed with an infinite trials formulation" -- This is extremely vague. Please make your assertion more direct and testable. Especially when this is the core motivation for the paper.
> > >
> > > We have shown that the objective functions of the infinite trials and the finite trials formulation of the convex RL problem are different (not equivalent). Thus, we cannot look at the infinite trials objective when we are actually addressing the finite trials objective.
> > >
> > > > " (3) stationary policies are in general sufficient" This also needs to be made more clear. You have not introduced a notion of stationarity, nor have you explicitly stated for what they are sufficient. Without this clarity inclusion of this statement in the introduction is meaningless.
> > >
> > > The definition of a stationary policy is reported in lines 102-103. We will also include a notion of stationary policy in the introduction following the Reviewer’s suggestion.
> > >
> > > > In the classic RL literature, if I am not mistaken, stationary usually means that the policy
> > > produces a value function which satisfies the bellman equations.
> > >
> > > In the RL/MDP literature, a stationary policy is an action selection strategy that is time-consistent, i.e., does not change with time. In convex RL we do not have the standard notion of the value function and Bellman equation.
> > >
> > > > All of this is predicated on the idea that the expectation is taken before the model is run. Which in general its not, and many standard RL algorithms function and have been analyzed in this regime.
> > >
> > > We do not understand the inquiry of the Reviewer here. Would they elaborate more on what is their concern?
> > > We further stress that it does not matter when the expectation is taken in standard RL, because the infinite trials and the finite trials formulations have equivalent objectives.

---

> > > > ### Author Response · Authors · 2022-08-01
> > > > **Authors Response 4/4**
> > > >
> > > > > "Thus, the extended MDP formulation and the resulting insight cast some doubts on the notion that the convex RL is not significantly harder than standard RL [31, 30]." -- Just state what the finding is. Your statement here is so conservative so as to be basically meaningless.
> > > >
> > > > The number of states of the extended MDP grows exponentially with the horizon $H$, which means that solving the finite trials convex RL problem with the extended MDP is not tractable. We are not reporting a formal reduction here, thus we do not have a sharp conclusion on the tractability of the finite trials convex RL problem. However, we conjecture that it is computationally harder to solve than the standard RL formulation. A formal verification of this conjecture would be a nice direction for future works.
> > > >
> > > > > Its extremely difficult to untangle what exactly is being stated by 5.2.
> > > >
> > > > In Section 5.2, we show how to cast a convex MDP into a partially observable MDP. Since the set of stationary policies are not sufficient for POMDPs, this relationship hints that stationary policies might be not sufficient for convex MDPs as well.
> > > > > Are you claiming that Theorem 5.1 is unique in providing sub-linear regret for the task of RL under general utilities in the presence of function approximation / inexact optimization / inexact expectations? It seems as though there must be result that are similar (e.g. provide sub-linear regret in the online mirror decent literature. Might look here (3)).
> > > >
> > > > We are not claiming anything like that. Th. 5.1 serves to prove that the finite trials convex RL problem is statistically tractable when the $\mathcal{F}$ function is known, because it exists at least one algorithm that achieves sub-linear regret. There might be other/better algorithms.
> > > > The reference that the Reviewer is mentioning addresses a different problem than the one we care about in this paper, as their algorithm is designed for an infinite trials formulation (see Eq. 2.1).
> > > >
> > > > > These are nice toy examples but if the point of this paper was to provide a better bridge between theory and practice please include an example which actually has a reasonably large state action space. Most problems that are used in practice e.g. autonomous vehicles or estimation of volatility usually have horizons and state-action spaces that are significantly larger then what is described here. Without a stronger, or clearer theoretical contribution I think more experiments illustrating your point would be helpful.
> > > >
> > > > This is a theoretical paper that is centered around a simple yet impactful finding: There is a mismatch between the infinite trials and the finite trials formulations of the convex RL problem. Since most of the methodologies for convex RL are designed for the infinite trials objective but address finite trials settings in practice, we believe this finding can have huge practical implications. We believe that an experimental analysis beyond the validation provided falls out of the scope of this theoretical paper. Moreover, we are not proposing a methodology here, so it is still unclear how to tackle finite trials convex RL problems in large settings. Instead, designing methodologies and providing empirical analysis would be an excellent matter for future work.
> > > >
> > > > > Code not included in supplemental material.
> > > >
> > > > We have now included in the supplementary material the code to run the numerical validation. Moreover, we will add a more detailed description of how to reproduce the experiments in the Appendix.

---

> > > > > ### Comment · Reviewer_CnKV · 2022-08-09
> > > > > **Follow up 4**
> > > > >
> > > > > Thanks for clarifying your relationship to prior work. And differentiating results that you have proven from conjectures that you are proposing. My only comment here is in response to:
> > > > > ```
> > > > > This is a theoretical paper that is centered ... Moreover, we are not proposing a methodology here, so it is still unclear how to tackle finite trials convex RL problems in large settings. Instead, designing methodologies and providing empirical analysis would be an excellent matter for future work.
> > > > > ```
> > > > > and again follows from my fundamental issue with this paper which is that, as a practitioner I am unsure as to what these results actually provide me that was not very clear from a practical perspective for some time. A single example of a popular IL / IRL algorithm (that is actually used practice) whose design could be changed based on this work to improve performance, would have been very much appreciated by anyone who cares about actually using the findings of this work in practice.

---

> > > > ### Comment · Reviewer_CnKV · 2022-08-09
> > > > **Follow up 3**
> > > >
> > > > I think you have addressed each of the questions I had here. My only comment is that the statement:
> > > > ```
> > > > Here we mean that the infinite trials convex RL formulation is computationally tractable, i.e., there exists an algorithm that computes the optimal policy in a polynomial number of steps when the MDP is known, and statistically tractable, i.e., there exists an algorithm that learns a slightly sub-optimal policy by taking a polynomial number of samples.
> > > > ```
> > > > is a bit misleading, as for most practical problems (which are rarely tabular) this is not really the case. Also to clarify the point made, and I think the fundamental issue with the paper at large:
> > > > ```
> > > > All of this is predicated on the idea that the expectation is taken before the model is run. Which in general its not, and many standard RL algorithms function and have been analyzed in this regime.
> > > > ```
> > > > I only meant to say that any practical instantiation of what the authors refer to as convex RL in settings like imitation learning or inverse reinforcement learning, already actively account for the mismatch that this work is predicated on. These practical algorithms do not have the guarantees that the authors provide obviously, but they also do not suffer from the specific types of degeneracy that are discussed in this work. I appreciate the interest, or possible need for discussion amongst the theoretical community, but it is very unclear what the practical community gains from these results, as they have been forced to deal with them for some time.

---

> > > ### Comment · Reviewer_CnKV · 2022-08-09
> > > **Follow up 2**
> > >
> > > Thank you for clarifying the use of equivalence and sufficiency. I also agree that a simple distinction between the papers which I mentioned is the linearity of the reward, though this is not unexpected considering that they are over 15 years old at this point. It was surprising to me that no other work in this area has extended inexact state-occupancy estimates to convex rewards. Such an analysis would obviously need to match your own,  as the thesis of your paper points out. Though because I have not interacted with this area, I am less familiar with the literature, and will give the authors the benefit of the doubt.

---

> > ### Comment · Reviewer_CnKV · 2022-08-09
> > **Follow up 1**
> >
> > Thanks for answering some of my questions. I have some final points below which I wanted to mention based on the author responses. I am satisfied with almost all of the responses in this section made by the authors except for the following statement.
> > ```
> > As the Reviewer noted, we only consider tabular settings in this paper. We will make this explicit in an updated version. However, we disagree that the tabular setting is not general: The tabular representation is the most general RL formulation, which admits all other settings as special cases.
> > ```
> > This is absolutely not true. Please explain how tabular settings generalize infinite state spaces or continuous state action spaces.

---

> ### Author Response · Authors · 2022-08-06
> **Is there any remaining concern?**
>
> We hope that our responses have addressed the Reviewer's concerns with sufficient clarity and detail.
>
> Could the Reviewer please let us know if there is any remaining concern to be addressed? We would like to use the rest of the discussion period to clarify any potential remaining issue.

---

> > ### Comment · Reviewer_CnKV · 2022-08-09
> > **Response Summary**
> >
> > Thanks for the thorough responses to my questions. As is stated below, my primary concern is exactly what I am supposed to take from this work as a practitioner that uses RL/IRL/IL. The results given here seem like a misunderstanding amongst the theory community as to what is done, or can actually be done in practice. Based on the interest from the other reviewers however, I will raise my score slightly. But I reiterate, from a practical perspective, this work offers no new information which can be clearly used to produce better algorithms to solve the class of problems it proposes to analyze.

---

### Official Review · Reviewer_EMAG · 2022-07-10

**Rating:** 8
**Confidence:** 4
**Soundness:** 4 excellent
**Presentation:** 3 good
**Contribution:** 3 good

**Summary:**

This paper analyzes the setting of convex reinforcement learning, where the objective is a function (mostly convex) of the state visitation distribution induced by an agent's policy.
The convex RL formulation as has been recently defined optimizes an objective that assumes access to the agent policy's state visitation distribution after executing this policy for an infinite number of trajectories. In practice, the objective is optimized using finite samples.

The paper argues that optimizing this infinite trials objective is not equivalent to optimizing the finite trials objective due to Jensen's inequality, and could lead to sub-optimal results. It also shows that the above mismatch does not exist for the setting of maximizing undiscounted cumulative rewards, due to the linear relationship between the state visitation distribution and the rewards vector in that setting.

Then, the paper further analyses the specific setting of a convex RL in a single trial. Here, the paper shows that casting the convex RL problem into an extended MDP where the state space is augmented to include all possible histories allows the problem to be cast into a linear reward setting, and thus allows it to be solved using finite trial RL settings. However, the policy space here scales exponentially with the horizon in this setting. The rest of the analysis considers the ramifications of instead casting this problem as a partially observable MDP (POMDP) or through the lens of online learning.

Finally, the paper backs up the analysis with some illustrative examples which optimize three objectives used in convex RL by optimizing the finite trials objective directly or instead optimizing the infinite trials objective with a finite number of trials. These examples seem to bear out the analysis in the paper.

**Questions:**

## Suggestions:
* While Section 5 adds some depth to the analysis of the paper, it also seems slightly off topic for the paper. Better motivation for section 5, and a greater preview of these results in the beginning of Section 5 as well as the introduction would be helpful.
* Citation number [30] from the paper has been published at a peer-reviewed conference. It would be better to use that citation [3] instead of ArXiv.

## Questions:
* Could ideas that try to characterize the full distribution induced by the agent's policy [1] and correct for the sampling error [2] be useful for optimizing the infinite trajectory objective with some off-policy correction?

### References:
[3] Zahavy, Tom, Brendan O'Donoghue, Guillaume Desjardins, and Satinder Singh. "Reward is enough for convex MDPs." Advances in Neural Information Processing Systems 34 (2021): 25746-25759.

**Limitations:**

Adequately

**Strengths And Weaknesses:**

## Strengths

* The paper's main result is fairly simple, straightforward, and impactful. It brings to light a discrepancy that might negatively affect applications of RL to convex objectives of the state visitation distribution.
* The analysis is clear, and the reason for the discrepancy pointed out is acceptable.
* The idea behind the illustrative examples is sound, and they are set up to validate the main idea of the paper.
* Section 5 adds in some depth to the analysis in the paper, and takes some steps to propose possible ways in which the finite trials objective might be optimized.

## Weaknesses

* While the idea behind the illustrative examples is well-intentioned, the way the results are presented leaves something to be desired. For example, in Figures 4 (d), (e), and (f), what is the x-axis?
* The text and Figure 4 caption contradict each other. Text: "optimal finite trials policy $\pi^\dagger$ and the optimal infinite trials policy $\pi^*$". Caption: "With $\pi^*$ we denote an optimal finite trial policy, with $\pi^\dagger$ an optimal infinite trial policy".
* The related work would be better rounded out by considering work in linear RL that has considered analyzing the full distribution induced by an agent's policy [1] and also work that has considered the impact of finite samples in such settings [2].

### References:
[1] Chandak, Yash, Scott Niekum, Bruno da Silva, Erik Learned-Miller, Emma Brunskill, and Philip S. Thomas. "Universal off-policy evaluation." Advances in Neural Information Processing Systems 34 (2021): 27475-27490.

[2] Pavse, Brahma, Ishan Durugkar, Josiah Hanna, and Peter Stone. "Reducing sampling error in batch temporal difference learning." In International Conference on Machine Learning, pp. 7543-7552. PMLR, 2020.

---

> ### Author Response · Authors · 2022-08-01
> **Authors Response**
>
> We want to thank the Reviewer for their positive feedback, and for deeming our work impactful and clear. We will revise the paper to improve the presentation of the illustrative examples and the motivation for Section 5 according to the Reviewer's suggestions. We provide below detailed answers to their questions.
>
> > In Figures 4 (d), (e), and (f), what is the x-axis?
>
> The x axes denote the value of the objective ($H(d^\pi)$, $\text{CVaR}_\alpha (\pi)$, $\text{KL} (d^\pi || d_E)$ respectively), while the y axes denote the probability of the outcome. We will make up for this typo, we are sorry for the confusion on the axes' labels.
>
> > Could ideas that try to characterize the full distribution induced by the agent's policy [1] and correct for the sampling error [2] be useful for optimizing the infinite trajectory objective with some off-policy correction?
>
> This is a great question, we thank the Reviewer for the interesting pointers, which we will surely mention in a final version of the paper.
>
> The work [1] considers a fairly different setting with respect to ours, as they address off-policy evaluation with performance metrics that go beyond the standard expectation of the return. Instead, we consider policy optimization problems in a convex RL setting, which cannot be modeled as a functional of the return distribution in general. However, the idea of employing importance sampling corrections to optimize a policy for the finite trials objective while deploying a policy optimized for the infinite trials objective is interesting, and certainly a matter for future works.
>
> The work [2] considers the issue of estimation in temporal difference learning with a finite amount of samples. Crucially, they tackle the standard RL formulation, where the infinite trials and the finite trials objective functions coincide. In our convex RL setting, the infinite trials and the finite trials objectives do not coincide even when they are estimated exactly. Thus, reducing the estimation error induced by the finite samples regime is not enough to solve the mismatch between the two objective functions.

---

> > ### Comment · Reviewer_EMAG · 2022-08-07
> > **Thank you for the Response**
> >
> > I thank the authors for their response, and their clarification of how the question this paper seeks to answer compares to approaches in off-policy evaluation as well as other approaches in RL literature that try to deal with sampling error.
> > I remain confident in my assessment of this paper.

---

### Official Review · Reviewer_J5z6 · 2022-07-11

**Rating:** 7
**Confidence:** 3
**Soundness:** 3 good
**Presentation:** 4 excellent
**Contribution:** 3 good

**Summary:**

The paper points out that the usual way convex RL is formulated implicitly assumes infinite trials, which can be a problem when learned policies are used in a finite number of trials (i.e., any application). The paper then introduces a formulation of convex RL that is appropriate for use with a finite number of trials, proves an upper bound on the approximation error introduced by optimizing the infinite trials objective instead of the finite trials objective, validates this bound empirically, and highlights other questionable assumptions from the convex RL literature.

**Questions:**

1. Could the authors better motivate Section 5.3 at the start of the section? At the end of the section it concludes that $\mathcal{F}$-specific algorithms will need to be developed for each of the specific instances of convex RL, but up until that point it was not really clear to me why we were doing this regret analysis.
1. Could the authors elaborate on why OPE-UCBVI is a principled algorithm for this setting? Is it near optimal? Is the regret analysis for this single algorithm actually a good representation of how hard the problem is?
1. How exactly were the policies in Section 6 arrived at? The text says they were obtained by optimizing the finite trial objective, but does not elaborate.
1. Can existing algorithms for optimizing the infinite trials convex RL objective be easily modified to optimize the finite trials version? It seems very unlikely, given the reduction of convex MDPs to extended MDPs and POMDPs. It would be good to emphasize this takeaway throughout the paper (at least in the conclusion), as it has the biggest implications for other researchers in the area; their algorithms are unlikely to transfer directly to the finite trials convex RL objective.
1. Is the optimal policy in Figure 4 (c, f) also non-Markovian?
1. Can the authors expand the discussion of related works [6] and [7] to clarify how the paper's contributions differ? A related works section is better if it not only summarizes related work but also explains how the current work is different/similar to the related work.

### Suggestions
- The caption for Figure 4 contradicts the text in the paragraph starting at line 268 about whether the optimal finite trials policy is $\pi^*$ or $\pi^\dagger$.
- It would be good to clarify the x-axes in Figure 4.
- Line 278: the text describes a reward vector "r", but the definition of $\mathcal{F}$ for risk-averse RL doesn't contain an "r" anywhere. Should "r" be $\lambda$ or vice versa?

**Limitations:**

The paper addressed a limitation of the bound on the approximation error, stating that it's not instance-dependent, as well as the limitation of the finite trials objective being less expressive in terms of policy orderings w.r.t. infinite trials convex RL.

**Strengths And Weaknesses:**

### Originality
To the best of my knowledge, the contributions in this paper are novel, but I am not an expert in convex RL.

### Quality
The paper seems very well-done; I couldn't find many issues with quality. I would've liked to see more explicit discussion of the implications of the proposed finite-trials convex RL objective, but there was a good amount.

### Clarity
The paper is extremely well-written and clear, with both the text and derivations being easy to follow. Actually a joy to read. This is very important for a paper pointing out a problem with the conventional way of doing things.

### Significance
The paper points out an issue with the current convex RL formulation, which includes several important sub-fields of RL, such as risk-averse RL, imitation learning, diverse skills discovery, constrained RL, etc. For this reason, the paper could be very significant.

---

> ### Author Response · Authors · 2022-08-01
> **Authors Response 1/2**
>
> We thank the Reviewer for their positive feedback and for praising the originality of the paper, the clarity of the presentation, and the significance of our contribution. We provide below detailed answers to their questions and suggestions.
>
> **Questions**
>
> > 1. Could the authors better motivate Section 5.3 at the start of the section?
>
> Thanks for the suggestion. We will report the motivations and results of Section 5.3 at the beginning of the section in an updated version. The analysis in Sec. 5.3 provides two interesting results: We show the importance of knowing $\mathcal{F}$ in the online setting, and we prove that the single trials convex RL problem is statistically tractable when $\mathcal{F}$ is known (i.e., there exists a learning algorithm achieving sub-linear regret).
>
> > 2. Could the authors elaborate on why OPE-UCBVI is a principled algorithm for this setting? Is it near optimal? Is the regret analysis for this single algorithm actually a good representation of how hard the problem is?
>
> Great question! The principled algorithms for online RL cannot be directly applied to the convex setting, as they assume to work with a linear objective, whereas other algorithms tackling the convex RL problem [e.g., 30, 31] are designed for the infinite trials setting. Instead, OPE-UCBVI is a good candidate, as the finite-trial convex RL problem can be formulated in the OPE setting with mild modifications. However, we do not claim that OPE-UCBVI is an optimal strategy for our setting, which means that our regret analysis is not necessarily tight. The goal of Section 5.3 is not to provide a comprehensive analysis of the statistical hardness of the problem, but rather a preliminary study of its tractability (for which we have a positive result when the function $\mathcal{F}$ is known). Establishing the fundamental statistical limits of the single-trial convex RL problem can be an interesting direction for future work.
>
> > 3. How exactly were the policies in Section 6 arrived at?
>
> We solved the extended MDP as described in Section 5.1 for the single-trial setting. We will clarify this in the text, thanks for pointing out the missing information.
>
> > 4. Can existing algorithms for optimizing the infinite trials convex RL objective be easily modified to optimize the finite trials version?
>
> We do not believe that previous methods for convex RL [e.g., 30, 31] can be easily applied to the finite trials setting. They generally solve a sequence of (standard) RL problems, from which they get a mixture of deterministic Markovian policies. Instead, the finite trials RL problem is akin to POMDPs (as mentioned in Section 5.2), and it likely requires fully non-Markovian policies in general. Anyway, a deeper study on the methodologies to apply to the finite trials convex RL setting is an interesting future direction.
>
> > 5. Is the optimal policy in Figure 4 (c, f) also non-Markovian?
>
> Yes, to match the expert's distribution $d_E$ in any trial, as it is required to maximize the single trials objective, we need a non-Markovian policy.
>
> > 6. Can the authors expand the discussion of related works [6] and [7] to clarify how the paper's contributions differ?
>
> The works [6, 7] tackle an online learning problem in MDPs with vectorial rewards, i.e., in which the reward function provides a vector of outcome instead of a scalar value. For this setting, they provide an analysis of the class of policies required, and they propose an algorithm based on UCRL2 that achieves sub-linear regret in the average reward formulation (i.e., infinite horizon).
>
> Similarities: Their vectorial-reward formulation allows to encode several convex/concave objectives that fall in the convex RL formulation (see Section 6 of [6]). E.g., in Section 6.2 they show how the pure exploration objective can be obtained by defining a convenient vectorial reward function with dimension S (number of states).
>
> Differences: The crucial difference is that they address an online learning problem in the average reward setting, whereas we look at the finite-horizon setting. In the average reward, the difference between finite trials and infinite trials settings fades away, because the chain asymptotically converges to the stationary distribution in any trial.
>
> Bottom line: We see our contribution and the work in [6, 7] as essentially complementary, with little to no overlap. If one cares about the average reward setting, we would point to their results. If one cares about the finite-horizon setting instead, we would point to our result showing the mismatch between the infinite trials and the finite trials convex RL setting.

---

> > ### Author Response · Authors · 2022-08-01
> > **Authors Response 2/2**
> >
> > **Suggestions**
> >
> > We thank the Reviewer for the useful suggestions.
> >
> > > The caption for Figure 4 contradicts the text in the paragraph starting at line 268
> >
> > Thanks for pointing this out! We will correct the caption: $\pi^\star$ denotes the optimal infinite trials policy, while $\pi^\dagger$ denotes the optimal finite trials policy.
> >
> > > It would be good to clarify the x-axes in Figure 4.
> >
> > The x axes denote the value of the objective ($H(d^\pi)$, $\text{CVaR}_\alpha (\pi)$, $\text{KL} (d^\pi || d_E)$ respectively), while the y axes denote the probability of the outcome. We will make up for this typo, we are sorry for the confusion on the axes' labels.
> >
> > > Line 278: the text describes a reward vector $r$, but the definition of F for risk-averse RL doesn't contain an $r$ anywhere. Should $r$ be $\lambda$ or viceversa?
> >
> > Yes, $r$ should be $\lambda$ instead, thanks for noticing.

---

> > > ### Comment · Reviewer_J5z6 · 2022-08-09
> > > **Response from reviewer J5z6**
> > >
> > > Thank you for your helpful response. I like the paper even more now.

---

### Official Review · Reviewer_eN8u · 2022-07-12

**Rating:** 7
**Confidence:** 4
**Soundness:** 3 good
**Presentation:** 3 good
**Contribution:** 3 good

**Summary:**

In this paper, the framework of convex Reinforcement Learning (RL) which covers applications such as imitation learning, and risk-averse RL is considered.  In vanilla RL, the idea of approximating the infinite trial objective by the finite counterpart works. However, the authors have established this assumption does not hold true in the convex RL setting. Further, they have proved and shown using experiments that optimizing an infinite trial objective instead of the finite one in convex RL may lead to a significant approximation error. The other contribution of the paper is that the authors have shown that other common assumptions in the convex RL literature such as computational tractability and sufficiency of stationary policies require re-investigation.

**Questions:**

However, there are a few concerns as follows:
1.	In the introduction, the authors have given an example of a financial application. Some more motivating examples for the single-trial case need to be highlighted to strengthen the motivation.
2.	The equivalence of the reward function in the extended MDP as in Section 5.1 and that of a finite-horizon convex MDP is not clear.
3.	Why do we need to encode $\mathcal{F}$ into a linear representation in Section 5.3, and why this particular linear representation is chosen in this paper? Can we make the approximation term in Theorem 5.1 better by choosing an alternative linear representation? The linear dependence of the approximation term on $N$ is a limiting factor. Can the OPE algorithm be adapted to the setting where no approximation is required and $\mathcal{F}$ can be used directly?
4.	Although the issues in convex MDP are well-investigated in this paper, the contribution in terms of the learning algorithm and associated analysis is limited.


**Limitations:**

Overall, this is the first paper that establishes that the infinite trial objective optimization does not lead to the optimal policy in the finite trial in convex RL, unlike the classical RL. However, there are concerns in terms of the extended MDP formulation and the linear dependence on $N$ because of the employed approximation. Therefore, the contribution in terms of the online learning mechanism seems to be limited.

**Strengths And Weaknesses:**

The paper is well-written and easy to follow. The contribution of the paper is significant as this is the first paper that establishes that the infinite trial objective optimization does not lead to the optimal policy in the finite trial in convex RL, unlike the classical RL.

---

> ### Author Response · Authors · 2022-08-01
> **Authors Response**
>
> We thank the Reviewer for their valuable feedback and for praising the significance of our contribution. We provide detailed answers to their questions below, which we hope will make them further appreciate our work.
>
> > 1. In the introduction, the authors have given an example of a financial application. Some more motivating examples for the single-trial case need to be highlighted to strengthen the motivation.
>
> We will include additional examples following the Reviewer's suggestion. We believe that anything that happens in the real world is single (or finite) trials at its core. This is true for robotic tasks, autonomous driving, medical treatments, and clinical trials, to name a few applications.
>
> > 2. The equivalence of the reward function in the extended MDP as in Section 5.1 and that of a finite-horizon convex MDP is not clear.
>
> By equivalence of the two problems, we mean that they have the same objective function and admit the same set of optimal policies. With some reworking on the objective function of the extended CMP $\arg \max_{\pi \in \Pi} r_\ell \cdot d^\pi$ (lines 213-214), we can write
>
> $$ r\_\ell \cdot d^\pi = \sum\_{s\_\ell \in \mathcal{S}\_\ell} r\_\ell (s\_\ell) d^\pi (s\_\ell) =  \sum\_{s\_\ell \in \mathcal{S}\_\ell} \mathcal{F} (d\_{s\_\ell}) \mathbf{1} (|s\_\ell| = H) p^\pi (d\_{s\_\ell}) = \mathbb{E}\_{d\_{s\_\ell} \sim p^\pi} [\mathcal{F} (d\_{s\_\ell}) ]
>  $$
>
> which is equivalent to the finite-trial objective $\zeta_1 (\pi)$ reported in equation 5.
> We will report this detailed derivation in a final version of the paper.
>
> > 3(1). Why do we need to encode F into a linear representation in Section 5.3, and why this particular linear representation is chosen in this paper?
>
> The linear representation of $\mathcal{F}$ is not a requirement of the problem setting, but rather a requirement of the OPE framework as presented in [5, Chatterji et al., On the theory of reinforcement learning with once-per-episode feedback, 2021]. The choice of the Bernstein polynomial for the linear representation is due to the approximation error, which is particularly amenable to analysis. We will clarify in the text that both the choices of the OPE framework and the Bernstein representation are essentially arbitrary. However, we note that the goal of Section 5.3 is to show
> - the importance of knowing the function $\mathcal{F}$ (more on this below);
> - and that the single-trial convex RL problem is statistically tractable when $\mathcal{F}$ is known.
>
> Instead, we leave a more comprehensive analysis of the fundamental statistical limits of the single trials convex RL setting to complement the reported preliminary results  as future work.
>
> > 3(2). Can we make the approximation term in Theorem 5.1 better by choosing an alternative linear representation? The linear dependence of the approximation term on N is a limiting factor.
>
> This is a great question! It is perhaps possible to improve the approximation error term of the regret, as we do not report a lower bound in this setting. However, we note that the multiplicative factor of $N$ is unavoidable when $\mathcal{F}$ is not known (see derivation between lines 493 and 494), and thus we would need an approximation error rate $O(N^{\alpha - 1}), \alpha < 1,$ to break the linear dependence on $N$. This can be achieved with sufficiently high-dimensional linear representations, but the pure learning regret term would become linear in $N$, as it also depends on the size of the representation.
>
> > 3(3). Can the OPE algorithm be adapted to the setting where no approximation is required and F can be used directly?
>
> Yes, definitely. We comment on this setting in lines 256-259: When the function $\mathcal{F}$ is known, we can achieve a sub-linear regret $O(\sqrt{N})$ for the online single trials convex RL problem.
>
> > 4. Although the issues in convex MDP are well-investigated in this paper, the contribution in terms of the learning algorithm and associated analysis is limited.
>
> We believe that shedding light on the mismatch between infinite trials and finite trials convex RL is already a crucial contribution. We leave as future work a more comprehensive analysis of the online learning problem in the finite trials convex RL setting.

---

> > ### Comment · Reviewer_eN8u · 2022-08-08
> > **Thank you for the response**
> >
> > I thank the authors for their response. The response clarifies the paper better. I am satisfied with the authors’ response. I have increased my review score accordingly.

---

### Author Response · Authors · 2022-08-01
**Authors Response: General Comment**

We want to thank the reviewers for their detailed feedback and insightful comments, which will allow us to improve the paper following their suggestions.
We are especially pleased that the reviewers found the contribution of our work to be "significant" (R1, R2), "novel" (R2), "impactful" (R3), and "a great way to move the field forward" (R4), while they praised the paper as generally "well-written and easy to follow" (R1), "very well-done, extremely well-written" (R2), "clear" (R3).
We hope that the detailed answers and clarifications we provide below can further increase their appreciation of our work and make an even stronger case for accepting this submission for publication.

---

### Author Response · Authors · 2022-08-06
**Any remaining concern?**

We want to thank again the reviewers for spending their time reviewing our paper and for providing useful feedback.

Could the reviewers please let us know if they are satisfied by our replies, or if there is anything that we failed to properly address in the response? We would like to use the rest of the discussion period to clarify any potential remaining issue.

---

### Meta-Review · Area_Chair_QyLf · 2022-08-26

**Recommendation:** Accept
**Confidence:** Certain

**Metareview:**

This paper provides new insights for convex reinforcement learning using finite vs. infinite trial objectives. Many of the reviewers felt this was a significant theoretical contribution that was presented clearly with the potential for significant theoretical impact. However, there were also concerns about the practicality of these insights: whether existing "real-world" applications of RL and IL were susceptible to the identified theoretical weaknesses of infinite trial objectives, and whether the experiments in this paper were sufficiently complex to demonstrate the advantages of the insights. The latter of these concerns was more widely shared among the reviewers. While I hope the authors might better address the practicality of their paper when revising their paper, I believe the theoretical contributions of the paper are sufficient for publication in NeurIPS and recommend acceptance.

**Award:**

No

---

### Decision · Program_Chairs · 2022-09-14

Accept